# Perovskites of the Tazheran Massif (Baikal, Russia)

**Eugene V. Sklyarov** [1,2,*] , **Nikolai S. Karmanov** [3] , **Andrey V. Lavrenchuk** [3,4] and
**Anastasia E. Starikova** [3,4]

1   Siberian Branch of the Russian Academy of Sciences, Institute of the Earth's Crust, 128 Lermontov st.,
    Irkutsk 664033, Russia
2   School of Engineering, Far East Federal University, 8 Sukhanov st., Vladivostok 690091, Russia
3   Siberian Branch of the Russian Academy of Sciences, V.S. Sobolev Institute of Geology and Mineralogy,
    3 Akad. Koptyuga st., Novosibirsk 630090, Russia; krm@igm.nsc.ru (N.S.K); alavr@uiggm.nsc.ru (A.V.L.);
    starikova@igm.nsc.ru (A.E.S.)
4   Department of Geology and Geophysics, Novosibirsk State University, 2 Pirogov st., Novosibirsk 630090,
    Russia
*   Correspondence: skl@crust.irk.ru; Tel.: +7-914-908-9408

**Abstract:** The paper provides details of local geology and mineralogy of the Tazheran Massif, which
was the sampling site of perovskite used as an external standard in perovskite U-Pb dating by
sensitive high-resolution ion microprobe (SHRIMP) and laser ablation inductively-coupled plasma
(LA–ICP–MS) mass spectrometry. The Tazheran Massif is a complex of igneous (mafic dikes,
syenite, nepheline syenite), metamorphic (marble), and metasomatic (skarn, calc–silicate veins)
rocks. Metasomatites are thin and restricted to the complex interior being absent from the margins.
Perovskite has been studied at four sites of metasomatic rocks of three different types: forsterite–spinel
calc–silicate veins in brucite marble (1); skarn at contacts between nepheline syenite and brucite
marble (2), and skarn-related forsterite–spinel (Fo-Spl) calc–silicate veins (3). Pervoskite from Fo-Spl
calc–silicate veins (type 1) is almost free from impurities (<1 wt.% in total: 0.06%–0.4% $REE_2O_3$,
0.10%–0.22% $Nb_2O_5$, ≤0.1% $ThO_2$). The U contents are from 0.1 to 1.9 wt.% $UO_2$ and are relatively
uniform in perovskites from the same vein but differ from vein to vein of this type. Perovskite from
the contact skarn (type 2) contains 1.5 to 4.5 wt.% $REE_2O_3$ but is poor in other impurities. Perovskite
grains from skarn-related Fo-Spl calc–silicate rocks (type 3) belong to two generations that differ in
REE, Nb, Th, Fe, and Na concentrations. Early-generation perovskites occurs as compositionally
homogeneous octahedral or cubic-octahedral crystals with contents of impurities higher than in
other varieties (3.6 wt.% $REE_2O_3$, 1.6 wt.% $Fe_2O_3$, 1.3 wt.% $Nb_2O_5$, 0.7 wt.% $ThO_2$, 0.6 wt.% $UO_2$,
and 0.6 wt.% $Na_2O$) but the lowest is at the respective site. Late-generation varieties show highly
variable impurity concentrations of 1.5 to 22.7 wt.% $REE_2O_3$, 0.4 to 8.4 wt.% $Nb_2O_5$, and 0.8 to 4.5%
$ThO_2$, while the perovskite component may be as low as 65%. In addition to the lueshite and loparite,
components, they contain $REEFeO_3$ and $Th_{0.5}TiO_3$ endmembers which have no natural analogs.

**Keywords:** perovskite; Tazheran Massif; skarn; calc–silicate veins; Ne–syenite; mafic dikes; Baikal;
Russia

## 1. Introduction

Perovskite is a widespread phase in $SiO_2$-undersaturated magmatic rocks (alkaline and Ne–syenite,
kimberlite, lamprophyre), and is a frequent mineral in metasomatic rocks at contacts with alkaline
intrusions (e.g., [1,2]). Along with zircon, baddeleyite, and other U-bearing minerals, perovskite
has been used lately as a U-Pb chronometer for dating kimberlites [3–5], carbonatites [6,7], and
lamprophyre [8]. Kramers and Smith [9] were the first to apply the perovskite chronometer to date

South African kimberlites by the classical U-Pb mass spectrometry, while Ireland et al. [10] were pioneers in ion-ion mass spectrometry (SHRIMP) with reference to high-U perovskite from the "Tazheran skarn deposit in the Lake Baikal area of eastern Siberia" [3,9]. The recommended 463 Ma age of the Tazheran perovskite was determined from $^{206}Pb/^{238}U$ by the ID-TIMS method in the early 1980s [11], and then the mineral was used many times as a standard for dating kimberlites from different areas worldwide, by SHRIMP, as well as by laser ablation inductively-coupled plasma mass spectrometry (LA–ICP–MS), and the classic TIMS method [3,5,12–14]. Yet, all cited publications lack geological and mineralogical details of the Tazheran deposit, which were reported in the book by Konev and Samoilov [15] published only in Russian, while the crystals used as reference material were sampled during international geological field trips. The mentioning of "skarn" would lead to thinking about a common metasomatic aureole around an intrusion in carbonate rocks, but the true setting of the deposit is much more complicated. Furthermore, perovskites are found in different lithologies, belong to various mineral assemblages, and are chemically inhomogeneous. This paper provides a synopsis of local geology and mineralogy of the Tazheran Massif, as well as mineral chemistry of perovskite.

## 2. Geological Setting

The Tazheran Massif is located within the Olkhon metamorphic terrane, which is part of an Early Paleozoic Baikal collisional belt (Figure 1) on the southern periphery of the Siberian craton [16]. The terrane is a mosaic of shear zones composed of various gneisses, amphibolites, calcitic, and dolomitic marbles, and quartzites derived from different protoliths ([17] and references therein) and diverse igneous lithologies, such as granites, syenites (including nepheline varieties), gabbro, and ultramafic rocks. There are three major igneous–metamorphic units (Figure 2): (1) large gabbro intrusions among amphibolite and carbonate rocks; (2) granitic, gabbro, and ultramafic intrusions among abundant gneiss and less widespread amphibolite and marble; (3) small granitic and gabbro intrusions among marble and gneiss. The terrane acquired this structure during its collision with the craton and related Early Paleozoic strike–slip faulting [18].

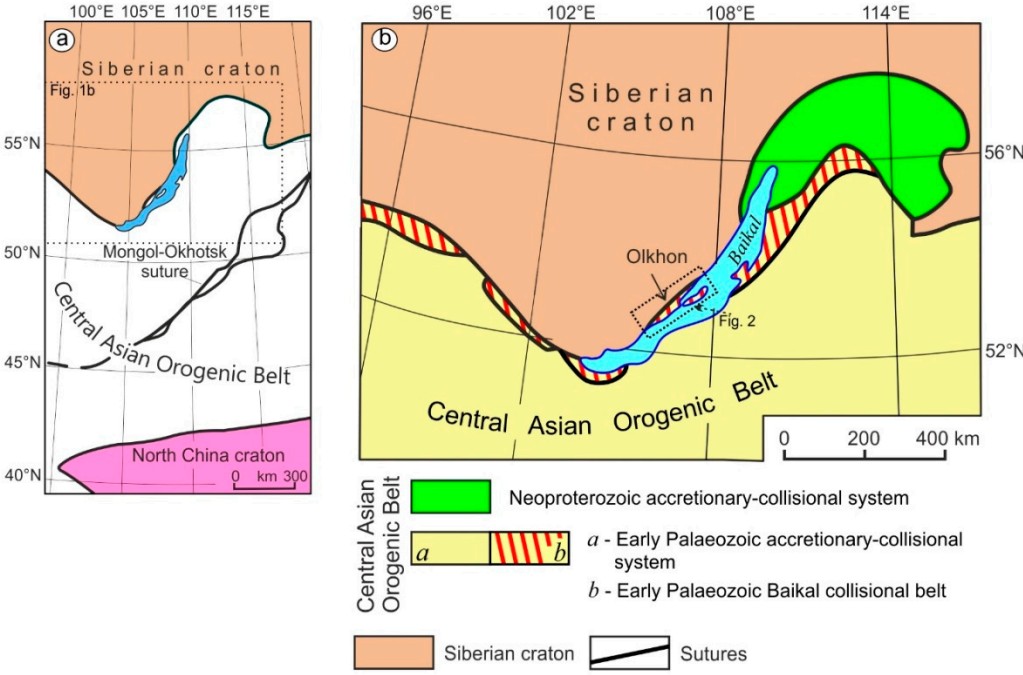

**Figure 1.** Simplified tectonics of Central Asia (**a**) and terranes in the Early Palaeozoic Baikal collisional belt of northern CAOB (**b**), modified after [16].

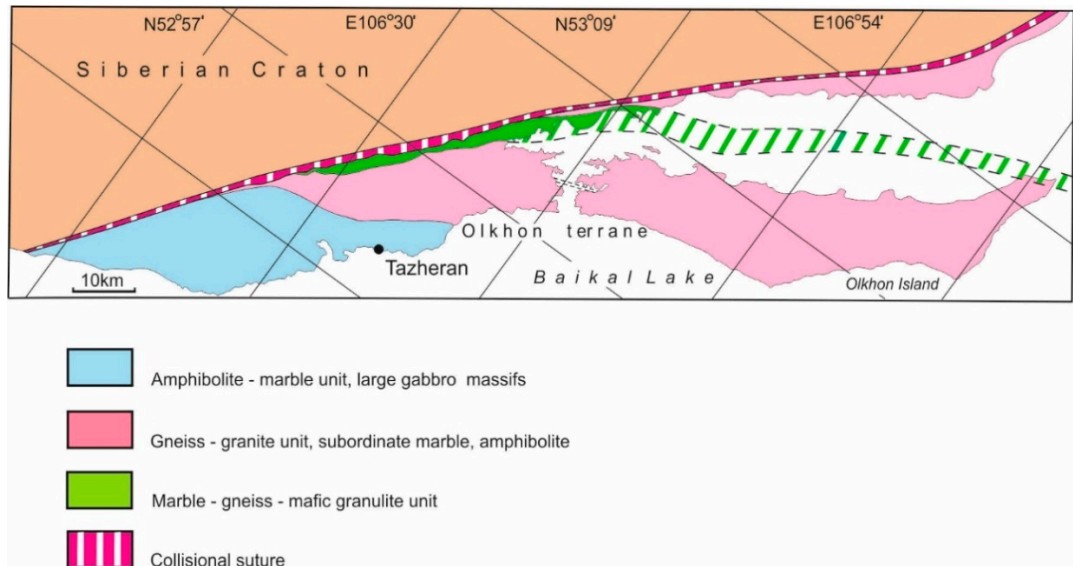

**Figure 2.** Simplified tectonics of the Olkhon terrane, modified after [16]: Tazheran Massif is shown as a black circle.

The Tazheran complex is an intricate mixture of igneous, metamorphic, and metasomatic rocks occupying ~6 km$^2$ on the shore of Lake Baikal (Figure 3). Igneous rocks are mostly gabbro–dolerite of phase 1 and foliated syenite; less abundant dikes of subalkaline microgabbro of phase 2; and minor nepheline syenite. Early gabbro—dolerite, dated at 470 Ma [19], has been almost fully converted to beerbachite: fine-grained massive or gneissic rocks with a mineral assemblage of Cpx+Opx+Pl+Ilm+Bt±Ol±Amp±Spl±Ti-Mag (abbreviations of minerals are after [20]). Microgabbro of phase 2 underwent metamorphism and has gneissic textures. As for nepheline syenite, it commonly occurs as linear or more complexly shaped veins varying in thickness from 20 cm to a few meters or even a few tens of meters. They are often too small to be resolvable even in 1:10000 map [21], and Figure 3 shows only areas where such veins are present. The rocks are crosscut by later pegmatite veins (not shown in Figure 3) that emplaced after the intrusion consolidation.

Metamorphic brucite marble occurs either as large blocks inside the complex or as veins crosscutting syenite and phase 2 microgabbro. Traditionally the marble has been interpreted as dolomitic xenoliths: It would convert to periclase marble at high temperatures and then transform to brucite marble as a result of periclase hydration [15]. However, judging by the presence of brucite marble veins that crosscut syenite and microgabbro, as well as by the absence of any relict periclase, the carbonate veins were likely injected [19,22] roughly coevally with the intrusion of nepheline syenite and phase 2 microgabbro dikes, for quite a long period of time within 454 and 469 Ma [23,24]. Identifying these carbonate veins as mantle carbonatite would contradict their mineralogy, chemistry [19], and stable isotope (O, C) signatures in rocks and minerals [25]. They rather can be called carbonatite-like dikes following Liu et al. [26] or crustal carbonatites, according to Wan et al. [27]. Their origin can be explained with the model of Lenz [28] for the melting of crustal carbonates upon contact with syenite magma in the presence of water. Further discussion of the issue is, however, beyond the scope of this paper.

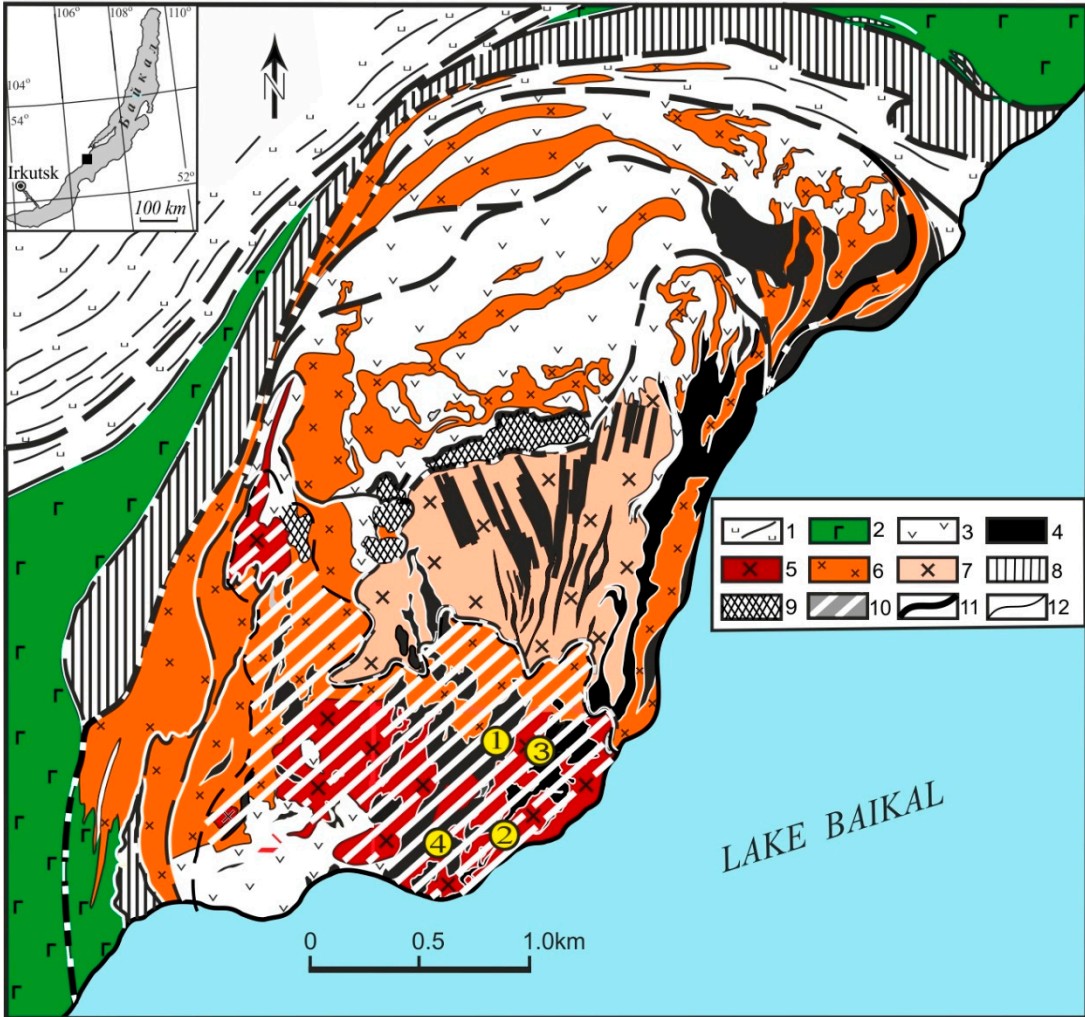

**Figure 3.** Geological sketch map of the Tazheran Massif, modified after [24]: 1–2 = country rocks: Amphibolite, silicate–carbonate gneiss (1); metamorphosed gabbro, monzogabbro, monzonite, syenite, 500 Ma (2); Tazheran Massif, 460–470 Ma (3–7): beerbachite after tholeiitic dolerite and gabbro (3), subalkaline gabbro and microgabbro (4), nepheline syenite (5), foliated (6) and massive (7) syenites; 8–10 = zones of metasomatic rocks of calcic (8), alkaline (9), and magnesian (10) types; 11 = synmetamorphic ductile detachment; 12 = geological boundaries.

Metasomatic rocks are widespread in the area and have magnesian, alkaline, or calcic affinities [24]. Magnesian varieties common to the southern part of the massif are either skarn and calc–silicate rocks at the contact between Ne–syenite and marble or calc–silicate veins in brucite marble. Calc–silicate veins are most often found near (but not immediately at) the contact with Ne–syenite but may also occur away from the latter. More details on magnesian metasomatites are provided in Section 4.

Alkaline metasomatites are Ti–fassaite–nepheline rocks and related garnet–melilite–diopside–wollastonite skarnoids (sometimes with minor nepheline or kalsilite) located in the middle of the complex where they coexist with dolomite-bearing calcite marble [29]. Alkaline metasomatites occur as bands at the contact of marble with phase 1 metadolerite or as 5 cm to 30 m fragments in the marble. Some contain 5 to 20 µm cubic crystals of perovskite.

Calcic metasomatic rocks are located among calcite marble around the Tazheran complex and follow linear amphibolite bodies (altered mafic dikes?). In some cases, they form thin zones within 10 cm (at the contact with marble or inside amphibolite) or up to 20 m thick bodies with relict amphibolite. They are typical medium-temperature garnet–pyroxene (±Amp, ±Ep, ±An) skarns free from perovskite.

When reading about perovskite as coming from the Tazheran skarn [3,5,12–14] one would imagine a typical skarn at a contact of a syenite intrusion with the host marble, but this is not the case. Even more, neither metasomatic rocks are present along the intrusion margins, nor does dolomite marble occur in their immediate vicinity. Few metasomatites (even those within intrusions) can be strictly classified as skarns, while the origin of other varieties requires special consideration in a separate paper.

## 3. Analytical Methods

The perovskite composition was studied in twelve thin polished sections and in fifty 1 to 3 mm crystals mounted on epoxy resin. Altogether, the analytical work included more than 400 analyses of perovskite samples from different sites, 2 to 30 points in each grain (55 analyses for site 1, 30 for site 2, 30 for site 3, and >300 for site 4); 120 analyses of accessories (Mg ilmenite, magnetite, geikielite, tazheranite, calzirtite, zirconolite, lueshite, baddeleyite, and uraninite–torianite); and 80 analyses of rock-forming minerals (forsterite, clinopyroxene, phlogopite, clinohumite, spinel, and calcite).

The analyses were performed at the Analytical Center for Multielement and Isotope Studies, Siberian Branch of the Russian Academy of Sciences (Novosibirsk), by scanning electron microscopy with energy dispersive X-ray spectroscopy (SEM-EDS) on a Tescan MIRA 3 LMU scanning electron microscope with an Oxford Inca Energy 450+/Aztec Energy XMax-80 and Inca Wave 500 microprobes. The operation conditions for EDS were: 20 keV beam energy, 1.5 nA beam current, and 20 s spectrum acquisition live time. Elements with atomic numbers $Z \leq 30$ (Zn) were determined using K-family X-ray lines; others were analyzed with L-family lines. Synthetic compounds and natural minerals have been used as standards: $SiO_2$ (O, Si), $BaF_2$ (F, Ba), $NaAlSi_3O_8$ (Na), $MgCaSi_2O_6$ (Mg, Ca), $Al_2O_3$ (Al), $Ca_2P_2O_7$ (P), $FeS_2$ (S), REE orthophosphates, and pure metals. Matrix correction was performed with the XPP algorithm as part of the built-in Inca Energy software. At these conditions, the lower detection limit was mostly within 0.2 to 0.3 wt.% ($Al_2O_3$, CaO, FeO, $K_2O$, MgO, MnO, $Na_2O$, $SiO_2$, $TiO_2$, $V_2O_3$) or rarely 0.5 to 0.8 wt.% or more ($HfO_2$, $Nb_2O_5$, PbO, $REE_2O_3$, SrO, $ThO_2$, $UO_2$, $ZrO_2$) in the case of spectral overlap. The metrological parameters of the SEM-EDS method are comparable with those of the classical electron microprobes with wave-dispersion spectrometry for major and minor components [30]: errors normally within 1 rel. % for major elements (C > 10–15 wt.%) and from 2 to 6 rel. % (no higher than 10 rel. %) for minor elements (C = 1–10 wt.%), and 20 to 30 rel. % for elements approaching the detection limit. Additionally, sites of epitaxially intergrown minerals were scanned at a 1000 $\mu m^2$ raster to determine average compositions. Most of the samples were analyzed at a small raster size (within 20 to 50 $\mu m^2$) to avoid effects of grain microtopography; the spot scanning mode was applied to objects less than 5 $\mu m$.

The perovskite mineral chemistry was also analyzed on a Camebax-Micro microprobe at a beam diameter of 2 to 3 $\mu m$, a beam current of 45 nA, an accelerating voltage of 20 kV, and a count time of 10 s per element. The following calibration standards were used: pyrope for Fe and Al; Mn-garnet for Mn; diopside for Mg and Ca; zircon for Hf, Zr, and Si; synthetic $LiNbO3$ for Nb; ilmenite for Ti; Cr-pyrope for Cr; synthetic $NaLa(MoO_4)_2$ for La; synthetic $CeF_3$ for Ce; albite for Na; Y-garnet for Y; Sr-glass for Sr; synthetic $UO_2$ for U; and synthetic $ThO_2$ for Th. EMPA data for Ti-Zr and other minerals were corrected for Si-Sr, Zr-Nb, Ti-Ba, Th-U, Fe-F, and Cr-Mn spectral overlaps. Lower detection limits for elements are (wt.%): 0.03 (CaO), 0.04 (FeO, SrO), 0.05 ($SnO_2$, $TiO_2$), 0.06 (MgO, $ThO_2$), 0.07 ($SiO_2$), 0.08 ($Nd_2O_3$, $ZrO_2$), 0.09 ($Pr_2O_3$, $UO_2$), 0.11 ($Na_2O$), 0.12 ($Ce_2O_3$, $Nb_2O_5$), 0.13 (BaO), 0.14 ($Al_2O_3$, $La_2O_3$), 0.16 ($Nb_2O_5$).

## 4. Geology and Mineralogy of Perovskite-Bearing Rocks

Perovskite commonly occurs in skarns at the contact between nepheline syenite and brucite marble or in forsterite–spinel (Fo-Spl) calc–silicate veins. We studied perovskite from metasomatic rocks of three types: forsterite–spinel calc–silicate veins in brucite marble (1); skarn at contacts between nepheline syenite and brucite marble (2), and skarn-related forsterite–spinel calc–silicate veins (3).

### 4.1. Contact Skarn and Calc–silicate Rocks

The Tazehran skarns are restricted to the contact between nepheline syenite and brucite marble but are absent from the contact of marble with foliated syenite or subalkaline gabbro. The skarns are most often 5 to 30 cm thick or occasionally reach a few meters. The skarn mineralogy consists of rock-forming forsterite, diopside, phlogopite, calcite, nepheline, less abundant amphibole, and accessory Mg ilmenite, magnetite, apatite, perovskite, baddeleyite, zirconolite, calzirtite, tazheranite, or rarely lueshite, and U-Th minerals. Out of several types of metasomatic skarn columns present in the Tazheran deposit, we consider a typical example from a well exposed and exhaustively documented outcrop (Figure 4). The outcrop displays several 0.5 to 20 m bodies of trachytic nepheline syenite in a small field of brucite marble among subalkaline gabbro, with 10 to 70 cm skarns at the contact of Ne–syenite and marble. Nepheline syenites appear to be boudinated, but their small bodies are oval- or drop-shaped and lack gneissic texture at the margins which rather have the same magmatic trachytic texture as in the center. The pattern resembles carbonate–silicate mingling, but the molten state of marble is unlikely. We hypothesize that carbonate was possibly viscoplastic and almost as viscous as syenite, which intruded low-viscosity carbonate and produced syenite "bubbles" in a carbonate matrix and reaction skarns at the silicate-carbonate contacts [22].

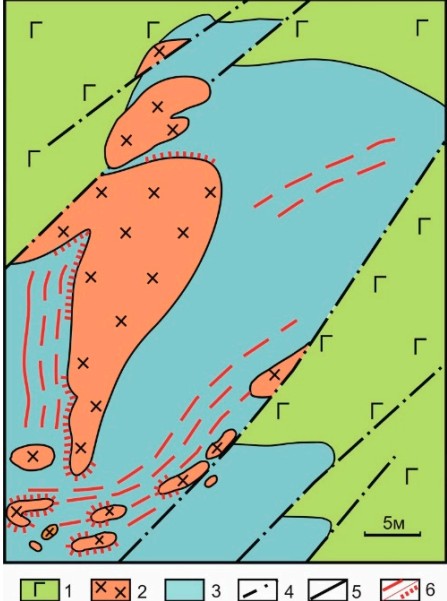

**Figure 4.** Detailed map of site 3, modified after [19]: 1 = amphibolite after microgabbro; 2 = nepheline syenite; 3 = brucite marble; 4 = ductile faults; 5 = contacts; 6 = calc–silicate veins and contact skarn.

The skarns comprise two zones between nepheline syenite and brucite marble: nepheline–pyroxene with phlogopite (i) and forsterite–spinel with phlogopite (ii). A large part of skarns are banded skarn (Figure 5A,B), but many are massive or breccia-like. Pyroxene in the nepheline–pyroxene zone has a uniform diopsidic composition (Table 1) with iron contents decreasing slightly away from syenite. Nepheline contains 15% to 19% kalsilite. Phlogopite has Mg# (Mg/(Fe+Mg)*100) = 91–93. The forsterite–spinel zone is 5 to 20 times thicker than the nepheline–pyroxene one (Figure 5A,B) and occasionally reaches 50 cm. It consists of alternated millimeter–centimeter thick carbonate and spinel–forsterite layers quasi-parallel to the contact. The forsterite component is as high as 91% to 92% (Table 1) and contains 0.3 to 0.65 wt.% MnO; $FeO_{tot}$ in spinel from the skarn margin is in a range of 10.5 to 13 wt.%. Perovskite rarely occurs in skarns; it has octahedral or cubic habits and crystal sizes mostly within 50 μm. The available U-Pb baddeleyite age of the skarns is 469 ± 5 Ma [24].

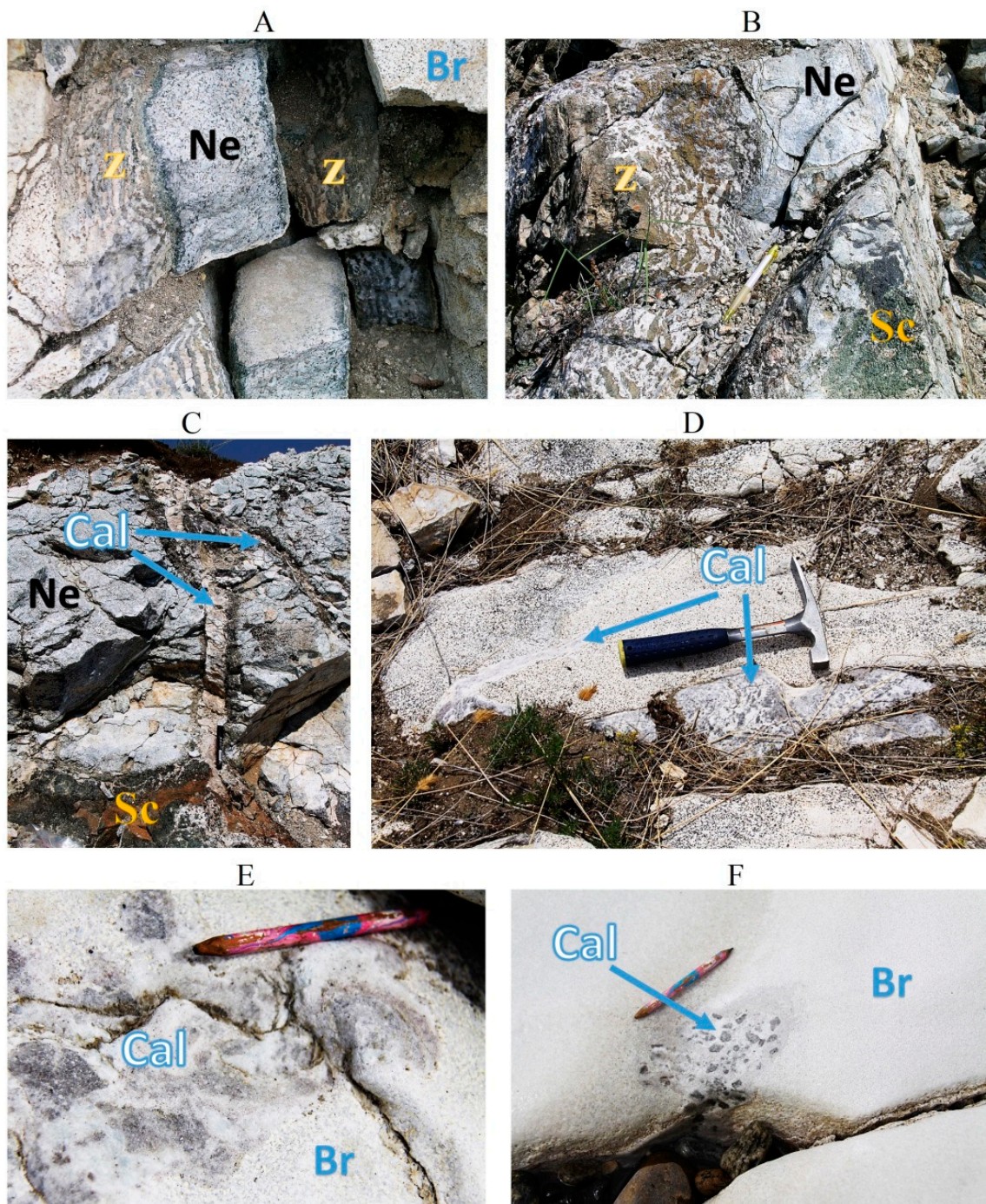

**Figure 5.** Photographs of perovskite-bearing metasomatic rocks: (**A**,**B**) Banded skarn (Z) at contacts of a thin vein of Ne–syenite (Ne); (**C**) calc–silicate veins (Cal) in Ne–syenite, with Fo-Spl-Cpx-Phl skarns (Sc) at the syenite-brucite marble (Br) contact; (**D**) Fo-Spl calc–silicate vein (Cal) with an offshoot in brucite marble (Br); (**E**,**F**) dismembered fragments of Fo-Spl calc–silicate vein (Cal) in brucite marble (Br).

**Table 1.** Selected analyses of spinel and forsterite of the Tazheran Massif.

| | Spinels | | | | | | | | Forsterite | | | | | | | |
|---|---|---|---|---|---|---|---|---|---|---|---|---|---|---|---|---|
| Site | 1 | | 2 | | 3 | | 4 | | 1 | | 2 | | 3 | | 4 | |
| SiO₂ | bdl | bdl | bdl | bdl | bdl | bdl | bdl | bdl | 41.91 | 42.04 | 42.34 | 42.5 | 41.03 | 40.96 | 41.3 | 41.4 |
| TiO₂ | 0.72 | 0.57 | 0.35 | 0.45 | bdl | bdl | 0.45 | 0.32 | bdl | bdl | bdl | bdl | bdl | bdl | bdl | bdl |
| Al₂O₃ | 68.40 | 69.25 | 69.26 | 69.49 | 66.16 | 65.96 | 66.59 | 67.26 | bdl | bdl | bdl | bdl | bdl | bdl | bdl | bdl |
| FeOtot | 3.47 | 3.20 | 1.45 | 1.33 | 11.28 | 13.04 | 7.42 | 6.02 | 1.02 | 1.10 | 0.30 | 0.35 | 8.04 | 8.75 | 5.54 | 4.95 |
| MnO | 0.04 | 0.04 | bdl | bdl | bdl | bdl | bdl | bdl | 0.06 | 0.06 | 0.17 | bdl | 0.48 | 0.35 | 0.32 | 0.23 |
| MgO | 27.40 | 27.25 | 28.04 | 28.03 | 22.26 | 21.25 | 24.56 | 25.35 | 56.82 | 56.33 | 56.59 | 56.8 | 50.65 | 50.35 | 52 | 52.1 |
| CaO | bdl | bdl | bdl | bdl | bdl | bdl | bdl | bdl | 0.06 | 0.02 | 0.00 | bdl | bdl | bdl | bdl | 0.25 |
| Total | 100.03 | 100.32 | 99.25 | 99.41 | 100.02 | 100.25 | 99.28 | 99.19 | 99.90 | 99.66 | 99.40 | 99.7 | 100.20 | 100.41 | 99.19 | 98.93 |
| Si | - | - | - | - | - | - | - | - | 0.989 | 0.994 | 1.000 | 1.001 | 0.997 | 0.995 | 1.002 | 1.003 |
| Ti | 0.012 | 0.010 | 0.006 | 0.008 | - | - | 0.008 | 0.006 | - | - | - | - | - | - | - | - |
| Al | 1.936 | 1.952 | 1.961 | 1.964 | 1.940 | 1.938 | 1.936 | 1.944 | - | - | - | - | - | - | - | - |
| Fe³⁺ | 0.037 | 0.028 | 0.027 | 0.020 | 0.060 | 0.062 | 0.047 | 0.044 | - | - | - | - | - | - | - | - |
| Fe²⁺ | 0.032 | 0.036 | 0.003 | 0.007 | 0.175 | 0.210 | 0.106 | 0.079 | 0.020 | 0.022 | 0.006 | 0.007 | 0.163 | 0.178 | 0.112 | 0.100 |
| Mn | 0.001 | 0.001 | - | - | - | - | - | - | 0.001 | 0.001 | 0.003 | - | 0.010 | 0.007 | 0.007 | 0.005 |
| Mg | 0.980 | 0.971 | 1.004 | 1.002 | 0.825 | 0.790 | 0.903 | 0.926 | 1.998 | 1.985 | 1.991 | 1.992 | 1.834 | 1.824 | 1.877 | 1.883 |
| Ca | - | - | - | - | - | - | - | - | 0.001 | 0.001 | - | - | - | - | - | 0.007 |
| Fo | | | | | | | | | 99.0 | 98.9 | 99.5 | 99.7 | 91.4 | 90.8 | 94.0 | 94.7 |

Note. bdl = below detection limit. 1,2,3,4 in line 2 = numbers of studied sites.

Separate skarn-related calc–silicate rock bodies (rather than layers in banded skarn) are restricted to a single site (Figure 6) where they coexist with skarns and consist of the same mineral assemblages as the latter but with 30% to 70% calcite. Calc–silicate rocks form small offshoots penetrating brucite marble in the western end of a nepheline syenite vein; calc–silicate veins crosscut nepheline syenite (Figure 5C) and thus postdates its crystallization. Calc–silicate rocks is most often a Fo-Spl variety, with up to 2 cm long forsterite crystals and 0.5 cm spinels; there is also pyroxene calc–silicate rocks with a breccia-like structure. Phlogopite sometimes makes small monomineralic lenses. Honey yellow forsterite contains a minor amount of iron (Fo = 94%–96%), while iron in Al spinel is from 5 to 10 wt.% (Table 1). Perovskite is very unevenly distributed and occurs as clusters of 1 to 10 mm octahedral or cubic–octahedral crystals in Fo-Spl calc–silicate rocks.

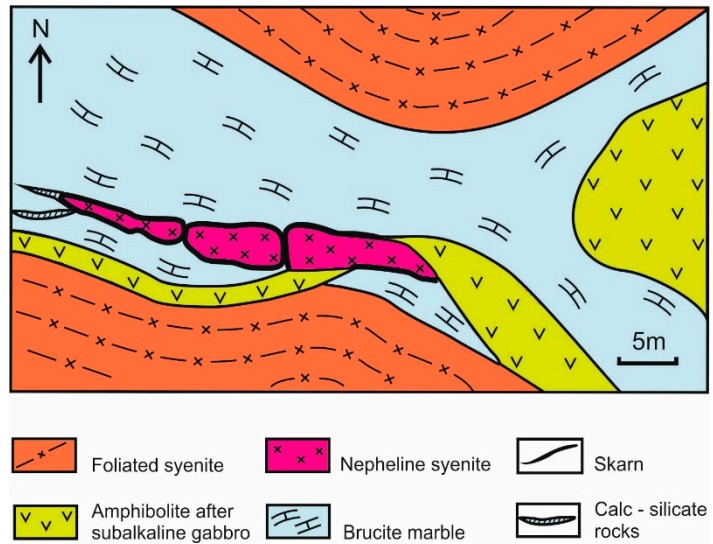

**Figure 6.** Detailed geologic map of site 4, modified after [24].

### 4.2. Calc–silicate Veins

Calc–silicate veins are widespread among brucite marble and are either forsterite or forsterite–spinel varieties. Forsterite calc–silicate rocks occur in abundance as thin (0.5–2 cm) parallel

veins, often in series, conformal to the marble–syenite contacts (Figure 4). In addition to calcite and forsterite (99.4–100 Fo), it may contain minor clinohumite and apatite but lacks other accessories.

Forsterite–spinel calc–silicate veins are relatively rare. Calc–silicate rocks form ≤10 m long veins of irregular shapes varying in thickness from 5 to 30 cm (occasionally up to 1 m), sometimes with thin offshoots into brucite marbles (Figure 5D). Some veins are totally dismembered in a viscoplastic flow of carbonate matrix and appear as fragments in brucite marble (Figure 5E,F). Many Fo-Spl calc–silicate rocks have breccia-like structures produced by calcite cementing of spinel-rich pieces (Figure 5D–F). The mineralogy is dominated by forsterite, Ti-bearing spinel and calcite while clinohumite and dolomite are less abundant. The accessories are perovskite, geikielite, Mg ilmenite, calzirtite, and tazheranite. Perovskite crystals are most often cubic or cubic-octahedral. Honey yellow forsterite (Fo = 99%–100%) is poor in iron, but Al spinel contains 5 to 10 wt.% FeO. They are the magnesian metasomatites (both vein and contact varieties) that bear the largest (up to 2 cm) cubic or cubic–octahedral perovskite crystals [15] and can make a suitable standard for U-Pb dating applications. However, perovskite is generally very unevenly distributed in calc–silicate rocks and most often occurs as sporadic clusters of 1 to 3 mm crystals.

## 5. Mineral Chemistry of Perovskite

Mineral chemistry was studied in detail for perovskites from four sites of the Tazheran calc–silicate rocks and skarn in rocks of three types. Two sites of Fo-Spl calc–silicate veins in brucite marble are located in an old trench (1 in Figure 3), where the perovskite crystals reported as reference materials for U-Pb dating (e.g., [3]) were most likely sampled, and in an outcrop on the shore of Lake Baikal (2 in Figure 3, details in Figure 5D–F). Skarns were sampled at the contact of nepheline syenite with brucite marble and in calc–silicate rocks: one between small drop-shaped syenite bodies and marble (3 in Figure 3, enlarged in Figure 4) and the other (4 in Figure 3) in Fo-Spl calc–silicate offshoots (Figure 6) on the extension of syenite.

Perovskites from the trench (1 in Figure 3) are especially pure, with impurities no more than 1 wt.% in total (Figures 7 and 8; Table 2): 0.10–0.19 wt% $Al_2O_3$, 0.12–0.21 wt% $Na_2O$, 0.1–0.2 wt% $ZrO_2$, 0.06–0.4 wt% $REE_2O_3$, 0.10–0.22 wt% $Nb_2O_5$, 0.16–0.2 wt% SrO, and up to 0.05wt% $UO_2$. This composition agrees well with previously published data [10,12], except for lower concentrations of iron (<0.1 wt.% against 1.23 wt.% $Fe_2O_3$) and strontium (~0.2 wt.% against 48 ppm SrO). Perovskites from Fo-Spl calc–silicate veins on the Baikal shore (2 in Figure 3) likewise contain minor amounts of impurities (Figures 7 and 8; Table 2), except for uranium (0.6-2.3 wt.% $UO_2$).

The skarn perovskites (site 3) differ in relatively high REE contents (Ce, La), from 0 to 1.8 wt.% $La_2O_3$ and 0.12 to 3.2 wt.% $Ce_2O_3$ (Table 2), while uranium is within 0.1 to 0.4 wt.% $UO_2$. Neodymium was not analyzed in these samples; in perovskites from other sites, it was higher than La but lower than Ce.

Perovskites from the skarn-related Fo-Spl calc–silicate offshoot (4 in Figures 3 and 6) show the largest ranges of element contents (Figures 7 and 8; Table 2). They contain up to 22.7% $REE_2O_3$, 8.5% $Nb_2O_5$, 4.3% $ThO_2$, 2.5% $Na_2O$, and 6% $Fe_2O_3$, but $UO_2$ is no higher than 1% and is often below the detection limit. All perovskites from this site fall into two distinct groups. More than 60 analyzed grains are euhedral and have relatively uniform compositions, with low contents of impurities (on average): 3.6 wt.% $REE_2O_3$, 1.6 wt.% $Fe_2O_3$, 1.3 wt.% $Nb_2O_5$, 0.7 wt.% $ThO_2$, 0.6 wt.% $UO_2$, and 0.6 wt.% $Na_2O$ (Figures 7 and 8). Rarely small grains with higher impurity contents appear in rims or at contacts of early perovskite with calcite inclusions (Figure 9A). The deviation from the perovskite composition *sensu stricto* is the greatest in irregularly shaped grains coated with epitaxially intergrown Mg ilmenite and magnetite (Figure 9B,C). These epitaxially intergrowths, some more magnetitic (Figure 9C) and others more ilmenitic (Figure 9D), envelope early perovskite with lesser amounts of impurities and late perovskite mostly restricted to small zones in rims or to separate grains of irregular shapes inside the magnetite-Mg ilmenite coat. In addition to perovskite, the coat encloses tazheranite, uraninite–torianite, nöggerathite–(Ce), and unidentified Ti-U-Th-Zr-Nb minerals (Figure 9E, Table 3). The compositional

heterogeneity in few perovskite grains (Figure 9F) is due to reactions with fluids rather than to earlier or later crystallization. Element contents in late perovskite are strongly variable: 1.5% to 22.7% $REE_2O_3$, 0.4% to 8% $Nb_2O_5$, and 0.8% to 4.8% $ThO_2$ (Figures 7 and 8; Table 2); uranium is within 1% $UO_2$ or often below detection limit.

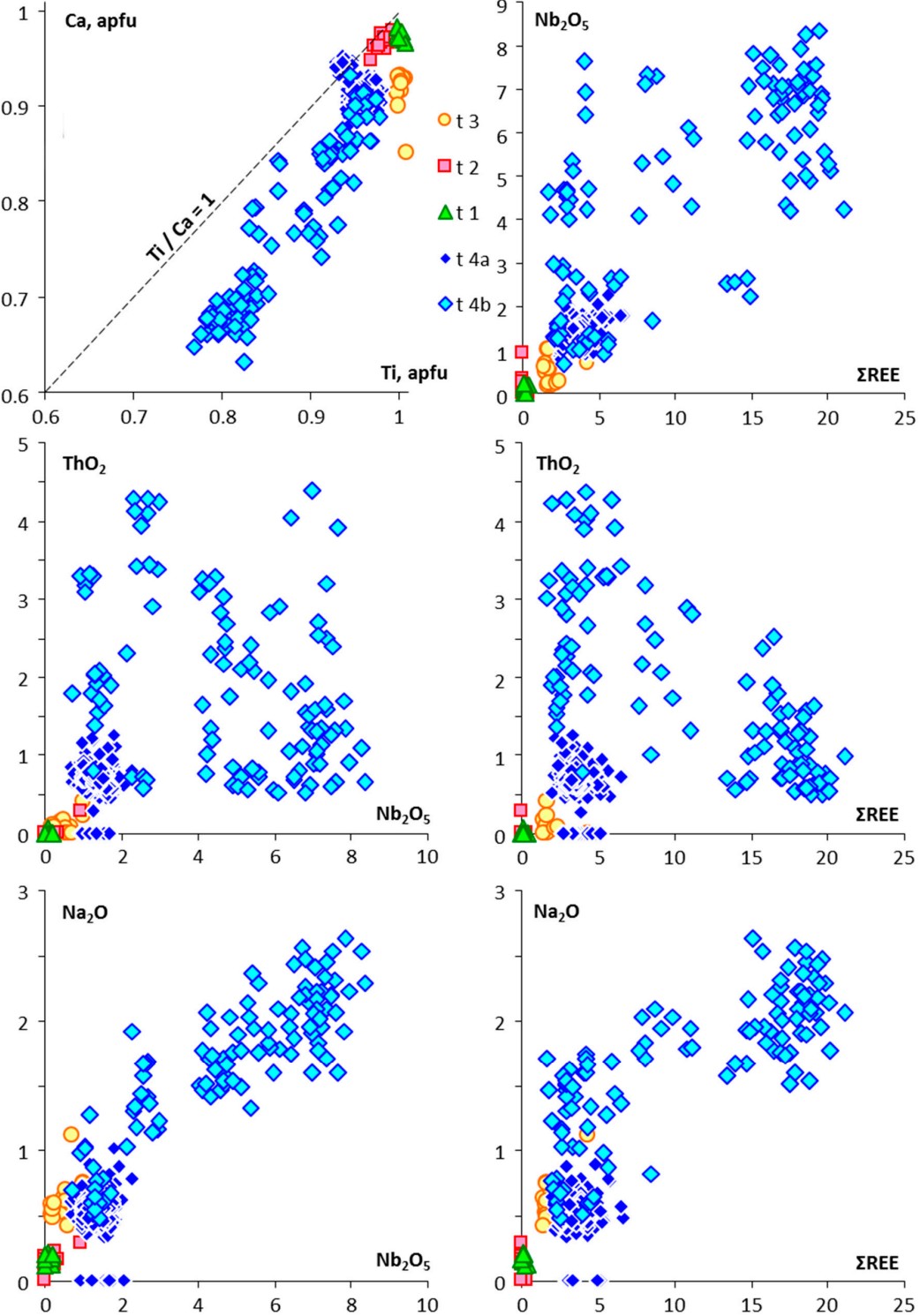

**Figure 7.** The Tazheran perovskites in major-oxide variation diagrams: Sites: 1 (green triangles), 2 (red squares), 3 (yellow circles), 4 (rhombs, dark blue for early generation and pale blue for late generation).

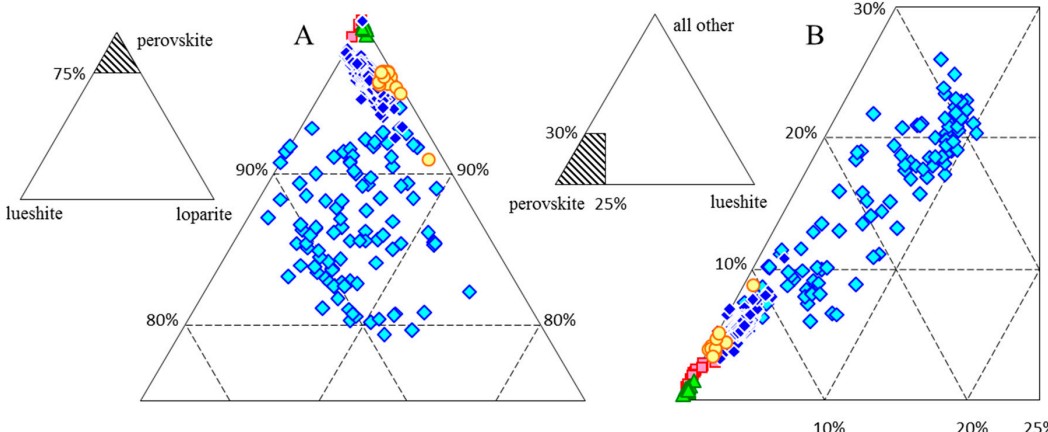

**Figure 8.** Compositional fields (mol%) in the ternary systems that include the recommended [31] lueshite–perovskite–loparite end-members (**A**) and all end-members (**B**): Symbols as in Figure 7.

**Table 2.** Selected analyses of perovskites of the Tazheran Massif.

| Site | 1 | | 2 | | | 3 | | 4 | | | | |
|---|---|---|---|---|---|---|---|---|---|---|---|---|
| | | | | | | | | early generation | | | late generation | |
| SiO₂ | bdl | bdl | bdl | bdl | 0.13 | 0.00 | 0.00 | 0.05 | 0.03 | 0.06 | bdl | bdl |
| TiO₂ | 58.19 | 58.31 | 57.13 | 57.13 | 55.53 | 58.08 | 57.58 | 53.30 | 53.38 | 53.19 | 54.54 | 52.31 |
| Al₂O₃ | 0.10 | 0.19 | 0.26 | 0.28 | 0.25 | 0.08 | 0.09 | 0.38 | 0.42 | 0.42 | 0.57 | 0.47 |
| Fe₂O₃ | 0.04 | 0.04 | 0.29 | 0.31 | 0.44 | 1.72 | 0.29 | 1.61 | 1.53 | 1.50 | 1.26 | 1.82 |
| V₂O₃ | bdl | bdl | 0.00 | 0.38 | 0.78 | bdl | bdl | bdl | bdl | bdl | bdl | bdl |
| MgO | 0.05 | 0.04 | 0.10 | 0.10 | 0.25 | 0.04 | 0.02 | bdl | bdl | bdl | bdl | bdl |
| CaO | 40.27 | 39.81 | 39.63 | 39.26 | 38.20 | 36.72 | 37.28 | 37.01 | 37.55 | 36.92 | 36.48 | 34.71 |
| SrO | 0.20 | 0.19 | bdl | bdl | bdl | 0.21 | 0.26 | bdl | bdl | bdl | bdl | bdl |
| Na₂O | 0.17 | 0.19 | bdl | bdl | 0.28 | 0.42 | 0.59 | 0.51 | 0.55 | 0.50 | 0.77 | 1.02 |
| Ce₂O₃ | 0.06 | 0.08 | bdl | bdl | bdl | 1.15 | 1.82 | 1.95 | 2.01 | 2.10 | 1.22 | 1.55 |
| La₂O₃ | 0.10 | 0.07 | bdl | bdl | bdl | 0.29 | 0.62 | 0.49 | 0.49 | 0.57 | bdl | bdl |
| Pr₂O₃ | n.a. | n.a. | bdl | bdl | bdl | n.a. | n.a. | 0.31 | 0.33 | 0.40 | bdl | bdl |
| Nd₂O₃ | n.a. | n.a. | bdl | bdl | bdl | n.a. | n.a. | 0.82 | 0.91 | 0.97 | 0.79 | 1.08 |
| Y2O3 | bdl | bdl | bdl | bdl | bdl | 0.07 | 0.11 | 0.46 | 0.30 | 0.33 | bdl | bdl |
| ΣREE₂O₃ | 0.15 | 0.15 | bdl | bdl | bdl | 1.51 | 2.55 | 3.57 | 3.74 | 4.04 | 2.01 | 2.63 |
| ThO₂ | 0.00 | 0.00 | bdl | bdl | 0.28 | 0.00 | 0.08 | 0.62 | 0.72 | 0.80 | 1.92 | 2.31 |
| UO₂ | 0.05 | 0.00 | 0.83 | 1.34 | 2.35 | 0.21 | 0.11 | 0.73 | 0.63 | 0.56 | 0.51 | 0.74 |
| Nb₂O₅ | 0.18 | 0.22 | bdl | bdl | 0.92 | 0.58 | 0.25 | 1.35 | 1.54 | 1.24 | 1.34 | 2.16 |
| ZrO₂ | 0.10 | 0.21 | 0.27 | 0.24 | bdl | bdl | bdl | bdl | bdl | bdl | bdl | bdl |
| Total | 99.66 | 99.34 | 98.48 | 99.02 | 99.65 | 99.54 | 99.09 | 99.43 | 100.24 | 99.43 | 99.40 | 98.17 |
| Si | - | - | - | - | 0.003 | - | - | 0.001 | 0.001 | 0.001 | - | - |
| Ti | 0.997 | 1.001 | 0.993 | 0.990 | 0.969 | 0.996 | 1.004 | 0.949 | 0.944 | 0.949 | 0.961 | 0.944 |
| Al | 0.003 | 0.005 | 0.007 | 0.008 | 0.007 | 0.002 | 0.002 | 0.011 | 0.012 | 0.012 | 0.016 | 0.014 |
| Fe³⁺ | 0.001 | 0.001 | 0.005 | 0.005 | 0.008 | 0.059 | 0.010 | 0.029 | 0.027 | 0.027 | 0.022 | 0.033 |
| V | - | - | - | 0.007 | 0.015 | - | - | - | - | - | - | - |
| Mg | 0.002 | 0.001 | 0.003 | 0.003 | 0.009 | 0.001 | 0.001 | - | - | - | - | - |
| Ca | 0.983 | 0.974 | 0.981 | 0.969 | 0.949 | 0.898 | 0.927 | 0.940 | 0.947 | 0.940 | 0.916 | 0.893 |
| Sr | 0.003 | 0.003 | - | - | - | 0.003 | 0.003 | - | - | - | - | - |
| Na | 0.007 | 0.009 | - | - | 0.013 | 0.019 | 0.027 | 0.023 | 0.025 | 0.023 | 0.035 | 0.048 |
| Ce | - | 0.001 | - | - | - | 0.010 | 0.015 | 0.017 | 0.017 | 0.018 | 0.010 | 0.014 |
| La | 0.001 | 0.001 | - | - | - | 0.002 | 0.005 | 0.004 | 0.004 | 0.005 | - | - |
| Pr | - | - | - | - | - | - | - | 0.003 | 0.003 | 0.003 | - | - |
| Nd | - | - | - | - | - | - | - | 0.007 | 0.008 | 0.008 | 0.007 | 0.009 |
| Y | - | - | - | - | - | 0.001 | 0.001 | - | - | - | - | - |
| Th | - | - | - | - | 0.002 | - | - | 0.004 | 0.003 | 0.003 | 0.010 | 0.013 |
| U | - | - | 0.004 | 0.007 | 0.012 | 0.001 | 0.001 | 0.003 | 0.004 | 0.004 | 0.003 | 0.004 |
| Nb | 0.002 | 0.002 | - | - | 0.010 | 0.006 | 0.003 | 0.014 | 0.016 | 0.013 | 0.014 | 0.023 |
| Zr | 0.001 | 0.002 | 0.003 | 0.003 | - | - | - | 0.005 | 0.003 | 0.004 | - | - |
| Total | 2.000 | 2.000 | 1.996 | 1.993 | 1.996 | 1.997 | 1.999 | 2.011 | 2.015 | 2.011 | 1.994 | 1.994 |

**Table 2.** *Cont.*

| Site | 4 | | | | | | | | | | | |
|---|---|---|---|---|---|---|---|---|---|---|---|---|
| | late generation | | | | | | | | | | | |
| SiO$_2$ | bdl | bdl | bdl | bdl | bdl | bdl | bdl | bdl | bdl | bdl | bdl | bdl |
| TiO$_2$ | 52.04 | 52.68 | 51.71 | 51.74 | 47.52 | 43.62 | 44.13 | 43.05 | 41.90 | 40.36 | 42.83 | 40.00 |
| Al$_2$O$_3$ | 0.45 | 0.45 | 0.53 | 0.57 | 0.49 | 0.74 | 0.76 | 0.70 | 0.74 | 0.57 | 0.62 | 0.47 |
| Fe$_2$O$_3$ | 2.14 | 2.40 | 2.60 | 2.75 | 2.65 | 4.10 | 3.35 | 3.53 | 4.33 | 5.16 | 4.32 | 4.55 |
| V$_2$O$_3$ | bdl | bdl | bdl | bdl | bdl | bdl | bdl | bdl | bdl | bdl | bdl | bdl |
| MgO | bdl | bdl | bdl | bdl | 0.41 | 0.46 | 0.61 | 0.75 | 0.81 | 0.83 | 0.53 | bdl |
| CaO | 34.20 | 33.40 | 32.97 | 33.71 | 32.59 | 28.47 | 26.63 | 26.54 | 25.12 | 24.58 | 24.44 | 22.93 |
| SrO | bdl | bdl | bdl | bdl | bdl | bdl | bdl | bdl | bdl | bdl | bdl | bdl |
| Na$_2$O | 1.29 | 1.27 | 1.17 | 1.70 | 1.74 | 1.79 | 1.93 | 2.06 | 2.53 | 2.13 | 2.36 | 2.20 |
| Ce$_2$O$_3$ | 1.96 | 3.28 | 2.83 | 1.70 | 2.78 | 6.17 | 8.82 | 6.79 | 8.73 | 10.08 | 10.85 | 11.97 |
| La$_2$O$_3$ | bdl | 1.30 | bdl | bdl | bdl | 1.88 | 2.42 | 2.10 | 2.80 | 2.42 | 3.62 | 5.42 |
| Pr$_2$O$_3$ | bdl | bdl | bdl | bdl | bdl | 0.83 | bdl | bdl | bdl | 1.23 | bdl | 1.04 |
| Nd$_2$O$_3$ | 1.00 | 1.03 | 1.46 | bdl | 1.39 | 2.30 | 3.52 | 3.36 | 4.25 | 4.68 | 3.90 | 4.29 |
| Y$_2$O$_3$ | bdl | bdl | bdl | bdl | bdl | bdl | bdl | bdl | bdl | bdl | bdl | bdl |
| ΣREE$_2$O$_3$ | 2.96 | 5.61 | 4.29 | 1.70 | 4.16 | 11.18 | 14.76 | 12.25 | 15.77 | 18.41 | 18.37 | 22.72 |
| ThO$_2$ | 4.29 | 3.32 | 3.41 | 3.02 | 4.04 | 2.81 | 1.95 | 3.07 | 2.39 | 1.50 | 0.83 | 0.93 |
| UO$_2$ | bdl | 0.62 | bdl | 0.85 | 0.67 | bdl | bdl | bdl | bdl | bdl | bdl | bdl |
| Nb$_2$O$_5$ | 2.32 | 1.16 | 2.40 | 4.65 | 6.44 | 5.89 | 5.85 | 7.47 | 7.51 | 6.88 | 5.41 | 6.64 |
| ZrO$_2$ | bdl | bdl | bdl | bdl | bdl | bdl | bdl | bdl | bdl | bdl | bdl | bdl |
| Total | 99.70 | 100.91 | 99.09 | 100.67 | 100.71 | 99.07 | 99.96 | 99.42 | 101.10 | 100.42 | 99.71 | 100.4 |
| Si | - | - | - | - | - | - | - | - | - | - | - | - |
| Ti | 0.935 | 0.943 | 0.935 | 0.915 | 0.863 | 0.830 | 0.841 | 0.823 | 0.801 | 0.787 | 0.831 | 0.836 |
| Al | 0.013 | 0.013 | 0.015 | 0.016 | 0.014 | 0.022 | 0.023 | 0.021 | 0.022 | 0.017 | 0.019 | 0.015 |
| Fe$^{3+}$ | 0.039 | 0.043 | 0.047 | 0.049 | 0.048 | 0.078 | 0.064 | 0.067 | 0.083 | 0.101 | 0.084 | 0.106 |
| V | - | - | - | - | - | - | - | - | - | - | - | - |
| Mg | - | - | - | - | 0.015 | 0.018 | 0.023 | 0.028 | 0.031 | 0.032 | 0.020 | - |
| Ca | 0.875 | 0.852 | 0.850 | 0.849 | 0.843 | 0.772 | 0.723 | 0.723 | 0.684 | 0.683 | 0.676 | 0.683 |
| Sr | - | - | - | - | - | - | - | - | - | - | - | - |
| Na | 0.060 | 0.058 | 0.055 | 0.077 | 0.082 | 0.088 | 0.095 | 0.102 | 0.125 | 0.107 | 0.118 | 0.119 |
| Ce | 0.017 | 0.028 | 0.025 | 0.015 | 0.024 | 0.057 | 0.082 | 0.063 | 0.081 | 0.096 | 0.102 | 0.122 |
| La | - | 0.012 | 0.000 | 0.000 | 0.000 | 0.018 | 0.023 | 0.020 | 0.026 | 0.023 | 0.035 | 0.056 |
| Pr | - | - | - | - | - | 0.008 | - | - | - | 0.012 | 0.000 | 0.011 |
| Nd | 0.009 | 0.009 | 0.013 | 0.000 | 0.012 | 0.021 | 0.032 | 0.031 | 0.038 | 0.043 | 0.036 | 0.043 |
| Y | - | - | - | - | - | - | - | - | - | - | - | - |
| Th | 0.023 | 0.018 | 0.019 | 0.016 | 0.022 | 0.016 | 0.011 | 0.018 | 0.014 | 0.009 | 0.005 | 0.006 |
| U | - | 0.004 | - | 0.005 | 0.003 | - | - | - | - | - | - | - |
| Nb | 0.025 | 0.013 | 0.026 | 0.049 | 0.070 | 0.067 | 0.067 | 0.086 | 0.086 | 0.081 | 0.063 | 0.083 |
| Zr | - | - | - | - | - | - | - | - | - | - | - | - |
| Total | 1.996 | 1.993 | 1.985 | 1.990 | 1.998 | 1.995 | 1.983 | 1.981 | 1.992 | 1.991 | 1.990 | 2.037 |

Note. bdl = below detection limit; n.a. = not analyzed; 1,2,3,4 in line 1 = numbers of studied sites.

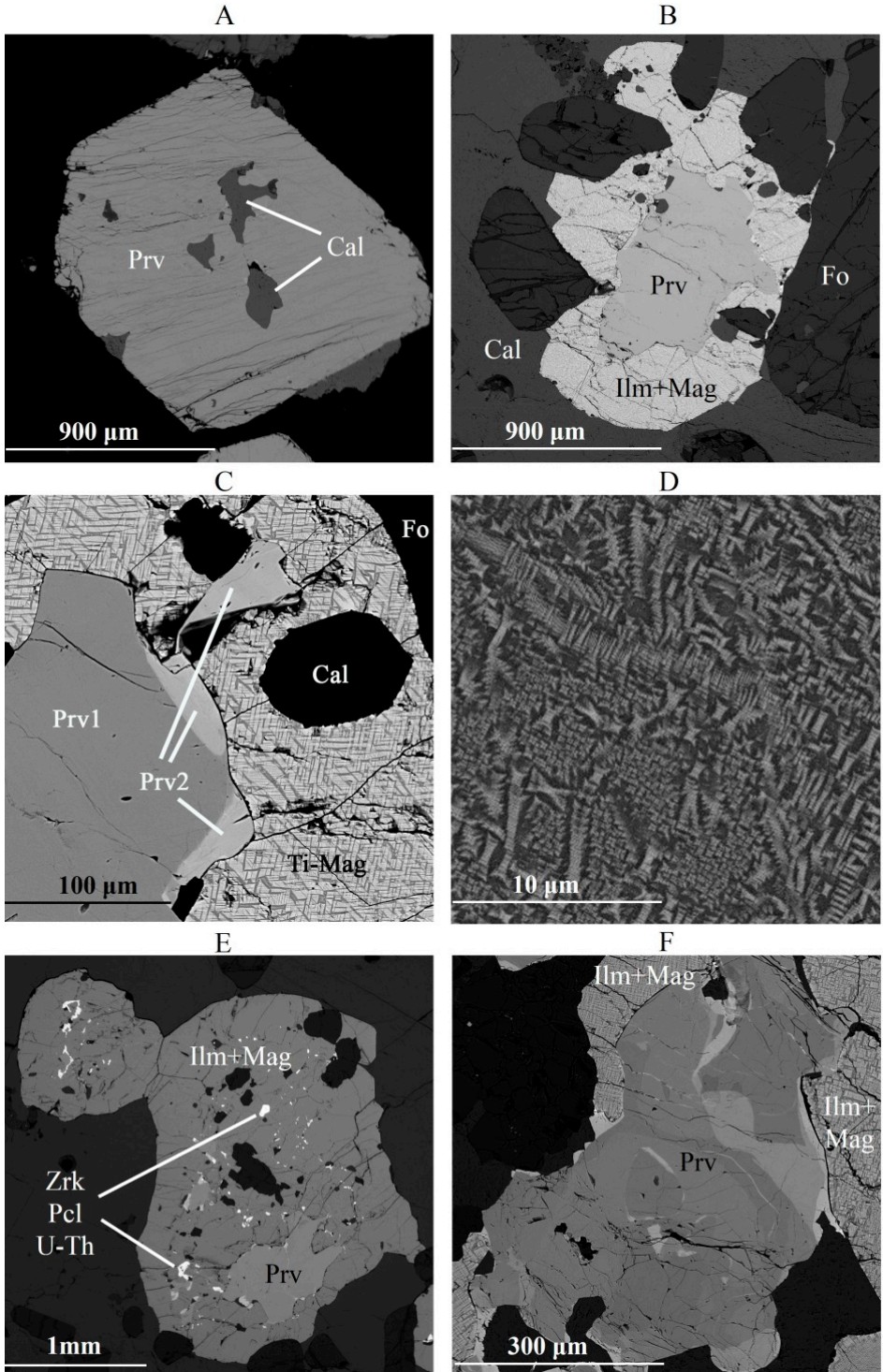

**Figure 9.** BSE images of perovskite from Fo-Spl calc–silicate rocks coexisting with skarns (site 4): (**A**) Compositionally homogeneous octahedral perovskite crystal with calcite inclusions; (**B**) irregularly shaped perovskite grain coated with epitaxially intergrown magnetite and Mg ilmenite; (**C**) a fragment of panel (B), with two generations of perovskite; (**D**) enlarged fragment of magnetite(bright)–ilmenite(dark) coat around a perovskite grain; (**E**) several perovskite grains in a magnetite–ilmenite coat with small inclusions of uraninite–torianite, nöggerathite–(Ce), and Ti-U-Th-Zr-Nb phases; (**F**) irregularly shaped and compositionally inhomogeneous perovskite grain in a magnetite–Mg–ilmenite coat. Abbreviations: zirkelite (Zrk), pyrochlore (Pcl), Ti-U-Th-Zr-Nb phases (U-Th), other minerals after [20].

**Table 3.** Selected analyses of Ti-Zr-U-Nb minerals of the Tazheran Massif.

| Site | 1 | 2 | 3 | | 4 | | | | |
|---|---|---|---|---|---|---|---|---|---|
| wt.% | Tzh | Tzh | Tzh | Zcl | Tzh | Th | Nb1 | Nb2 | Nb3 |
| $SiO_2$ | bdl | bdl | bdl | 0.12 | bdl | bdl | bdl | bdl | 0.30 |
| $TiO_2$ | 17.49 | 16.7 | 17.18 | 38.17 | 17.08 | bdl | 18.88 | 18.97 | 18.47 |
| $Al_2O_3$ | bdl | bdl | bdl | bdl | bdl | bdl | 0.30 | bdl | bdl |
| FeO | 0.13 | 0.19 | 0.28 | 2.53 | bdl | bdl | 5.92 | 1.30 | 1.93 |
| MnO | bdl | bdl | bdl | 0.09 | bdl | bdl | bdl | bdl | bdl |
| MgO | 0.11 | bdl | 0.05 | 0.15 | bdl | bdl | 2.24 | bdl | 1.76 |
| CaO | 12.01 | 12.23 | 11.64 | 13.12 | 11.78 | bdl | 7.74 | 15.38 | 4.98 |
| SrO | bdl | bdl | bdl | bdl | bdl | bdl | bdl | bdl | 0.72 |
| $Na_2O$ | 0.05 | bdl | 0.09 | 0.08 | bdl | bdl | bdl | 0.69 | bdl |
| $Ce_2O_3$ | 0.06 | bdl | 0.08 | 0.22 | bdl | bdl | 4.83 | 2.27 | 2.60 |
| $La_2O_3$ | bdl | bdl | bdl | bdl | bdl | bdl | 1.04 | bdl | bdl |
| $Nd_2O_3$ | bdl | bdl | bdl | bdl | bdl | bdl | 4.14 | 1.66 | 1.43 |
| $UO_2$ | bdl | 1.83 | bdl | 2.53 | 0.56 | 29.53 | 3.93 | 4.34 | 4.45 |
| $ThO_2$ | 0.03 | bdl | 0.19 | 0.90 | 0.00 | 64.21 | 4.73 | 21.79 | 20.50 |
| $ZrO_2$ | 67.35 | 66.68 | 68.30 | 36.70 | 69.16 | 0.00 | 28.83 | 2.44 | 2.49 |
| $HfO_2$ | 1.59 | 0.8 | 0.78 | 0.30 | 0.99 | bdl | bdl | bdl | bdl |
| PbO | bdl | bdl | bdl | bdl | bdl | 3.68 | bdl | 1.48 | 1.39 |
| $Nb_2O_5$ | bdl | bdl | 1.44 | 2.98 | bdl | bdl | 15.02 | 24.10 | 23.26 |
| Total | 98.82 | 98.43 | 100.03 | 98.19 | 99.57 | 97.42 | 97.59 | 94.41 | 84.27 |

Note. bdl = below detection limit. 1,2,3,4 in line 1 = numbers of studied sites. Tzh = tazheranite; Zcl = zirconolite; Th = thorianite; Ng = nöggerathite-(Ce); Nb1, Nb2 = not identified Ti-Ca-U-Th-Zr-Nb minerals.

## 6. Discussion

The mineral chemistry of the Tazheran perovskites from different sites turns out to be more variable than in the pure perovskite *sensu stricto* mentioned by Kinny et al. [3], with 1200 to 6300 ppm U. Perovskites from three sites (1,2,3) have quite uniform compositions. Those from Fo-Spl calc–silicate veins in brucite marble are the most proximal to perovskite *sensu stricto* (Figures 7 and 8): they have minor contents of REE, Th, Nb, and other impurities. The uranium concentrations are, however, different (up to 0.05 wt% U in site 1 and 1.3 wt.% $UO_2$ on average (up to 2.3 wt.%) in site 2). Perovskites from different Fo-Spl calc–silicate veins of type 1 differ uniquely in U while other impurities vary insignificantly. The skarn perovskites show higher REE, Nb, and Th enrichment at relatively low and uniform U contents (0.1–0.4 wt.% $UO_2$). Perovskites from sites 1, 2, 3 plot the most closely to pure perovskite in all diagrams (Figures 7 and 8).

Unlike these, perovskites from the skarn-related Fo-Spl calc–silicate rocks of site 4 are less homogeneous and have relatively high impurity contents (Fe, Na, REE, Nb, and Th), while the range of U contents is narrow. Early perovskites are more homogeneous and contain lesser amounts of impurities, but late varieties show large ranges of Fe, Na, REE, Nb, and Th and lower percentages of the perovskite end-member, down to 67% (Figure 8). As shown by calculations of end-members according to Locock and Mitchell [32], the shares of the lueshite, loparite, and $REEFeO_3$ end-members may reach, respectively, 10.5%, 14%, and 15%, while other end-members are rare to absent. The higher the impurity contents, the more the deviation from the line of constant Ca/Ti (Figure 7).

The contents of impurities are related in an intricate way. Niobium shows no correlation with REE and Th, with largely scattering points in diagrams (Figure 7), but it distinctly correlates with Na at >2 wt.% $Nb_2O_5$ (Figure 7). However, this correlation (which records the presence of the lueshite component) disappears at low $Nb_2O_5$. REE show correlation with Th (strong negative) and Na (weak positive) only as they reach notable percentages above 5 wt% (Figure 7). Each separate grain of late perovskite is most often quite uniform in chemistry, but sometimes there are some variations in Th and REE, even within individual grains.

The origin of skarn at the contact of nepheline syenite with brucite marble and skarn-related forsterite–spinel calc–silicate rocks appears quite obvious. Perovskite and other minerals in the hosts

of both types uptake Ti, REE, Nb, U, and Th from syenite melts. These elements become accumulated in late fluids that release from skarns and form offshoots in marbles. Nb, REE, and Th enrichment of second-generation perovskites in skarn-related calc–silicate rocks correspond to the composition trend typical of late metasomatism reported, for instance, for Ne–syenite, including that in the Khibiny mountains [33]. The successive gain by perovskite of Na, Th, Nb, and REE from fluids released from Ne–syenite melts is consistent with low contents of these and other elements in the Tazheran Ne–syenite [17]. On the other hand, the origin of spinel–forsterite calc–silicate veins in brucite marbles remains open to discussion. It may be due to the release of fluids or fluid–melt mixtures upon a reaction of Ne–syenite with carbonates at large depths and their subsequent injection into shallower crust. Going farther into this discussion would require considering many other features of the Tazheran complex, which is beyond the scope of this paper.

The compositional heterogeneity of many Tazheran perovskites makes their use for U-Pb dating questionable. Originally, they were the coarse cubic crystals from site 1 (trench) that were used as standard. Later, amateur stone collectors have depleted the deposit, the trench was filled back, and a very limited amount of suitable reference material (>1 mm grains) has been left. Perovskites are quite abundant at site 4, but some of them are compositionally and structurally inhomogeneous. The anhedral perovskites in magnetite–ilmenite coats contain late-generation REE-Nb-Th-rich perovskite grains, and other U phases hardly can be a good standard. However, the euhedral crystals are compositionally uniform and rarely belong to the late generation, while U contents, though being higher than in the currently used standard, are not very high (0.3–0.8 wt.% UO2).

## 7. Conclusions

The Tazheran Massif is a complex structure that results from multiple intrusions of syenites, mafic dikes, and Ne–syenites, as well as injections of carbonates, associated with strike–slip tectonics [21]. Metasomatism appears in the interior of the Massif but lacks from its margins.

The Tazheran deposit comprises three main types of perovskite-bearing metasomatic rocks: forsterite–spinel calc–silicate veins in brucite marble (1); skarns at the contact of nepheline syenite with brucite marble (2) and skarn-related forsterite–spinel calc–silicate rocks (3). The accessories in metasomatites of all types are perovskite, geikielite, Mg ilmenite, baddeleyite, apatite, calzirtite, tazheranite, magnetite, or less often zirconolite, and uraninite–torianite, as well as lueshite in a few cases.

Perovskite from Fo-Spl calc–silicate veins (type 1) bears the least amount of impurities (0.06–0.4 wt.% $REE_2O_3$, 0.10–0.22 wt.% $Nb_2O_5$, ≤0.1 wt.% $ThO_2$) and from 0.1 to 1.9 wt.% $UO_2$. The contents of $UO_2$ in perovskites are similar within each vein but differ between veins.

REE contents in the contact skarns (type 2) are notably higher (1.5–4.5 wt.% $REE_2O_3$) while concentrations of other impurities are low. Perovskites from Fo-Spl calc–silicate rocks related to skarns (type 3) are much more heterogeneous. There are early and late perovskite generations that differ markedly in the contents of REE, Nb, Th, Fe, and Na. The early compositionally homogeneous octahedral and cubic–octahedral perovskite crystals have higher concentrations of impurities than in the above varieties (3.6 wt.% $REE_2O_3$, 1.6 wt.% $Fe_2O_3$, 1.3 wt.% $Nb_2O_5$, 0.7 wt.% $ThO_2$, 0.6 wt.% $UO_2$, and 0.6 wt.% $Na_2O$), but they are the lowest at the respective site. The late-generation perovskites have variable concentrations of impurities and lower percentages of the perovskite end-member (down to 67%). In addition to the lueshite, and loparite components, they contain a $REEFeO_3$ and $Th_{0.5}TiO3$ endmember, which has no natural analogs.

**Author Contributions:** E.V.S. made geological study, collected samples, interpreted the data, prepared tables and figures, wrote the manuscript; N.S.K. performed analytical study, wrote the part of the manuscript related to analytical methods; A.V.L. made geological study, collected samples, prepared some figures; A.E.S. made geological study, collected samples, prepared some figures and tables.

**Funding:** Work has been carried out as part of government assignment at IEC SB RAS and IGM SB RAS, Ministry of Science and Higher Education of the Russian Federation. Analytical studies were partly supported by grant 18-17-00101 from the Russian Science Foundation.

**Acknowledgments:** We wish to thank T. Perepelova for assistance in manuscript preparation. Thoughtful comments by Academic Editor and two anonymous reviewers on this manuscript are gratefully acknowledged.

**Conflicts of Interest:** The authors declare no conflict of interest.

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
