# Peer review of "Perovskites of the Tazheran Massif (Baikal, Russia)"

_minerals, doi:10.3390/min9050323_

Round 1

Reviewer 1 Report

The manuscript is very confusing. The interpretation of data does not meet the standards for publishing at Minerals. Both terms and discussion are presented in a very inconsistent way. For examples, skarn is sometimes understood as the rock at the contact zone between nepheline syenite and marble (in the abstract); sometimes is understood to include all rocks from the contact zone to the surrounding rocks (nepheline syenite and marble as well). For the authors, fosterite-spinel calciphyre veins belong also to skarn. Therefore, although the study samples (perovskite) are actually from 3 different geneses (skarn, calciphyre vein in marble, and calciphyre vein in nepheline syenite) from the Baikal area (Russia), the title misleads to only one genesis (skarn).

The abstract is not clearly written, what is the overall purpose, the methods, and the main findings of the study.

The authors do not provide how many samples totally they used for their chemical analysis, how many samples for each genesis/each site of study.

There is only the comparison of samples but the reason for differences (probably genetic/environmental) between them is not explained. There is no explanation for the existences of impurities, neither (why they are there, in which lattice sites, which substitution mechanism, etc). A comparison between chemical compositions of different samples without any explanation for it does not bring much scientific meaning as the fact is, there are always differences, especially when the samples are from different geneses.

The authors also try to make argument to perovskite standards from other studies (for example the one used by Kinny et al – paper in Russian, and I have no access to this paper). However, if the authors did not analyze these standards, they have nothing to make argument even the standards are from the same deposit with their samples.

I would recommend to reject the manuscript.

Author Response

Dear reviewer, thank you for comments. My reply is in attached file.

Best regards,

Prof. E. Sklyarov

Reviewer 2 Report

Comments on minerals-482458 entitled ‘Perovskites of the Tazheran Skarn Deposit (Baikal, Russia)’ coauthored by Sklyarov et al., submitted to Minerals

Sklyarov et al. present a solid data on geology, and mineral chemistry of perovskite from the Tazheran deposit in the Baikal Lake area, Russia. They describe two main types of perovskite-bearing metasomatic rocks: 1) vein forsterite-spinel calciphyre in brucite marble; 2) skarns at the contact of nepheline syenite with brucite marble and related vein forsterite-spinel calciphyre, and point out that perovskite crystals from the former contains the least amount of impurities, suitable for U-Pb dating, whereas those from the latter are much more heterogeneous, thus may be ideal for the age dating. Based on this, the publication of this manuscript (ms) will raise a broad interest from mineralogical and geochronological communities, given few minors are addressed.

The nature of brucite carbonate veins (lines 85-87) and marbles need discussed, which appear to be carbonatitic in origin if the description as presented in the ms. Otherwise, rewording the description is required.

Few terms need revised in the text (see details in the edited PDF ms).

Figure 2: why do not use the same color as in Figure 1?

Figure 7: (a) Why do not plot Ti vs. Ca on the basis of apfu that may better portray the data than on wt.% plot?

Figure 9: Check the description on habit of perovskite crystasl, e.g. the image not indicative of cubic-.

Table 2: Check this table for some inconsistency with the text (e.g., up to 23% REE2O3), and correct it.

A number of minors are picked up and marked on the edited PDF ms for authors’ consideration when a revision is made to improve the presentation of this paper.

Author Response

Dear reviewer, thank you very much for your comments. We tried to correct the manuscript according to your comments. Details in attached file.

Best regards,

Prof. Eugene Sklyarov
